# Photoprotection Differences between Dominant Tree Species at Mid- and Late-Successional Stages in Subtropical Forests in Different Seasonal Environments

**DOI:** 10.3390/ijms23105417

**Published:** 2022-05-12

**Authors:** Wei Lin, Zhengchao Yu, Yanna Luo, Wei He, Guanzhao Yan, Changlian Peng

**Affiliations:** Guangzhou Key Laboratory of Subtropical Biodiversity and Biomonitoring, Guangdong Provincial Key Laboratory of Biotechnology for Plant Development, School of Life Sciences, South China Normal University, Guangzhou 510631, China; 13527044032@163.com (W.L.); yuzhengchao199301@163.com (Z.Y.); 2019022497@m.scnu.edu.cn (Y.L.); hfhew813@126.com (W.H.); guanzhao996@163.com (G.Y.)

**Keywords:** subtropical forest, dominant tree species, successional stage, photoprotection, seasons

## Abstract

Plants growing in subtropical regions are often affected by high temperature and high light in summer and low temperature and high light in winter. However, few studies have compared the photoprotection mechanism of tree species at different successional stages in these two environments, although such studies would be helpful in understanding the succession of forest communities in subtropical forests. In order to explore the strategies used by dominant species at different successional stages to cope with these two environmental conditions, we selected two dominant species in the mid-successional stage, *Schima superba* and *Castanopsis chinensis*, and two dominant species in the late-successional stage, *Machilus chinensis* and *Cryptocarya chinensis*. The cell membrane permeability, chlorophyll fluorescence, chlorophyll content, and a few light-protective substances of these dominant species were measured in summer and winter. The results show that in summer, the young leaves of dominant species in the mid-successional stage showed higher anthocyanin content and superoxide dismutase (SOD) activity, while those in the late-successional stage showed higher flavonoid and total phenolic content, total antioxidant activity, non-photochemical quenching (NPQ), and carotenoid/chlorophyll (Car/Chl) ratio. In winter, young leaves of dominant species in the mid-successional stage were superior to those in the late-successional stage only in terms of catalase (CAT) activity and NPQ, while the anthocyanin, flavonoids, and total phenol content, total antioxidant capacity, and Car/Chl ratio were significantly lower compared to the late-successional stage. Our results show that the dominant species in different successional stages adapted to environmental changes in different seasons through the alterations in their photoprotection strategies. In summer, the dominant species in the mid-successional stage mainly achieved photoprotection through light shielding and reactive-oxygen-species scavenging by SOD, while the antioxidant capacity of trees in the late-successional stage mainly came from an increased antioxidative compounds and heat dissipation. In winter, the dominant species in the mid-successional stage maintained their photoprotective ability mainly through the scavenging of reactive oxygen species by CAT and the heat dissipation provided by NPQ, while those in the late-successional stage were mainly protected by a combination of processes, including light shielding, heat dissipation, and antioxidant effects provided by enzymatic and non-enzymatic antioxidant systems. In conclusion, our study partially explains the mechanism of community succession in subtropical forests.

## 1. Introduction

Light and temperature are two important environmental factors affecting plant growth. These will be a negative impact on the normal growth of plants if the light is too strong or too weak. The effect of excessive light on plants is mainly caused by an imbalance between the light energy absorbed by the photosynthetic system and the light energy utilized. In a high-light environment, incomplete utilization of the light energy absorbed by plants leads to the generation of excess light energy, and then the phenomenon of photoinhibition occurs [1]. The stress of weak light on plants is mainly caused by the limitation of CO_2_ assimilation due to stomatal closure [2,3]. In a low-light environment, fixation of CO_2_ will be limited and the regeneration of NADP^+^ through the Calvin cycle will be reduced if the light energy absorbed by the plant is disproportionate to the magnitude of stomatal closure. Therefore, an excessive photosynthetic electron transport chain will induce the production of reactive oxygen species (ROS).

Similarly, high or low temperature will also affect the growth of plants. High temperature damages the chloroplast structure and reduces the content of photosynthetic pigments by disrupting the structure of enzymes involved in their synthesis [4,5]. Low temperature affects photosynthesis by limiting stomatal opening and reducing the activity of related photosynthetic enzymes [6]. No matter what kind of environmental stress (high light, low light, high temperature, or low temperature), the most fundamental effects are that it induces the production of excess light energy of the photosynthetic mechanism, causes the accumulation of unused and potentially harmful excitation energy on the photosynthetic membrane, and finally transmits the excitation energy to O_2_ to form reactive oxygen species [7]. Reactive oxygen species have a variety of destructive effects on plants. On the one hand, they can lead to the degradation of chloroplast D1 protein and block its synthesis [8,9], thus inhibiting photosynthesis. On the other hand, ROS can directly attack the lipid layer of the cell membrane, causing peroxidation of the lipid and increasing the permeability of the cell membrane [10]. Binding of malondialdehyde (MDA), a membrane lipid peroxidation product, to proteins and nucleic acids can lead to further cellular damage [11]. Compared with mature leaves, which have mature photosynthetic systems, the incomplete photosynthetic system of young leaves makes them more susceptible to various types of oxidative damage due to environmental influences.

In order to cope with environmental stresses, plants have developed different defense systems, among which the non-enzymatic and enzymatic antioxidant systems are two important forms of defense. The non-enzymatic antioxidant system is mainly composed of small molecular antioxidants, including flavonoids, total phenols, anthocyanins, etc. Flavonoids are widely found in plants and belong to the secondary metabolites. Increased flavonoid content in plants can be found in various environmental stresses conditions, such as excess solar radiation [12,13], temperature [14,15], water [16], etc. Currently, flavonoids found in plants mainly play an antioxidant role [17,18,19,20]. In addition, part of the antioxidant capacity in plants also depends on the synthesis of phenolic compounds. For example, it was found that in *Rubus ellipticus*, the scavenging ability of phenolic extracts to 1,1-diphenyl-2-picryl hydrazyl (DPPH) is better than that of other antioxidants [21]. As part of the plant stress defense mechanism, the enzymatic antioxidant systems usually respond to various stresses, such as temperature [22,23], high light [24], heavy metal [25], and water [26], whose components mainly include superoxide dismutase (SOD), catalase (CAT), peroxidase (POD), ascorbate oxidase (AOX), and glutathione reductase (GR) [27,28]. Non-photochemical quenching (NPQ) is a defense mechanism independent of the enzymatic and non-enzymatic antioxidant systems, and is an important photoprotective method for plants to eliminate excess light energy by thermal dissipation [29]. NPQ is dependent on the lutein cycle and can harmlessly consume excess energy captured by photochemical devices, alleviating the effects of excessive excitation energy on chloroplasts [30].

In the face of environmental stress, plants will flexibly adopt different defense strategies. For example, a study by Zhu et al. [31] showed that young leaves of *Acmena acuminatissima* can be protected by enhancing heat dissipation in summer, but they can enhance their photoprotective potential in winter by increasing anthocyanins content. The antioxidant system of rice grown under blue and red light is weakened, while the ability of heat dissipation through NPQ is enhanced [32]. Under high temperature, the antioxidant enzyme system of *Chlorella pyrenoidosa* is activated to scavenge overproduced reactive oxygen species [33]. These studies suggest that plants tend to strengthen certain defense strategies in response to different environmental stresses.

In subtropical regions, plants suffer from high temperature and high light stress in summer and low temperature and high light stress in winter. The question is what defensive measures will plants take in the face of different environmental stresses. In order to clarify the defense strategies of tree species at different successional stages in subtropical evergreen broad-leaved forests against different stress environments in summer and winter, two dominant tree species in the mid-successional stage and two dominant tree species in the late-successional stage were selected in this study. Their cell membrane permeability, chlorophyll fluorescence, chlorophyll content, anthocyanin content, flavonoid content, total phenolic content, total antioxidant capacity, and antioxidant enzyme activity were determined.

## 2. Results

### 2.1. Relative Membrane Leakage

In order to determine the degree of damage in dominant tree species in different successional stages under different environmental stresses, we measured the cell membrane leakage of young and mature leaves in different seasons. In both summer and winter, mature leaves had higher relative membrane leakage compared to young leaves (Figure 1), and the relative membrane leakage of young leaves of dominant species in the mid-successional stage was higher than that of dominant species in the late-successional stage.

### 2.2. Chlorophyll Fluorescence

In studying the photosynthesis of plants, it is common to characterize the degree of photoinhibition by Fv/Fm. In this study, young leaves had lower Fv/Fm values than mature leaves in both summer and winter (Figure 2A). In summer, the Fv/Fm values were significantly higher for young leaves of the dominant species in the mid-successional stage than in the late-successional stage, while in winter, the values were significantly higher in the late-successional. Contrary to Fv/Fm, in summer, NPQ was significantly lower for young leaves of the dominant species in the mid-successional stage than in the late-successional stage in summer (Figure 2B). In winter, NPQ of the young leaves increased in the mid-successional stage and decreased in the late-successional stage, and was significantly higher in the mid-successional stage.

### 2.3. Pigment Content

The chlorophyll content of mature leaves was significantly higher than that of young leaves (Figure 3A). In summer, the chlorophyll content of the young leaves of the dominant species in the mid-successional stage and the late-successional stage was almost equal. The content of chlorophyll in young leaves of the dominant species in the mid-successional stage was slightly higher than those in the late-successional stage in winter, but there was no statistical difference. Carotenoids showed an opposite trend to chlorophyll. The content of carotenoids was significantly lower in young leaves of the dominant species in the mid-successional stage than the dominant species in the late-successional stage in summer, but almost equal in winter (Figure 3B). Whether in summer or winter, the Car/Chl was significantly higher in young leaves of the dominant species in the late-successional stage than the dominant species in the mid-successional stage (Figure 3C).

Young leaves of the dominant species at different successional stages showed opposite color changes in different seasons. Young leaves of the dominant species in the mid-successional stage turned red in summer, and those in the late-successional stage turned red in winter. This change was due to the accumulation of anthocyanins. Thus, young leaves of dominant species in different successional stages had different anthocyanin accumulation patterns in different seasons. In summer, young leaves of dominant species in the mid-successional stage accumulated more anthocyanins than those in the late-successional stage (Figure 4A), while the former had a higher content of anthocyanins than the latter in winter. Anthocyanins were hardly accumulated in mature leaves.

In summer, flavonoids showed a different accumulation pattern from anthocyanins; the content of flavonoids in young leaves of dominant species in the late-successional stage was significantly higher than in those of dominant species in the mid-successional stage (Figure 4B). Similarly, the content of flavonoids was significantly higher in young leaves of dominant species in the late-successional stage than the mid-successional stage in winter. Compared with summer, the content of flavonoids in young leaves of dominant species in both stages was lower in winter.

In summer, the total phenolic content was significantly higher in young leaves of dominant species *Cryptocarya chinensis* in the late-successional stage than the other three tree species, and the total phenol content of the young leaves of the dominant species *Machilus chinensis* in the late-successional stage of was the lowest among all tree species. In general, the total phenolic content was higher in young leaves of dominant species in the late-successional stage than in the mid-successional stage (Figure 4C). In winter, the total phenolic content was significantly higher in young leaves of dominant species in the late-successional stage than in the mid-successional stage.

### 2.4. Total Antioxidant Capacity

The total antioxidant capacity of all tree species decreased in winter compared with summer. Whether in summer or winter, the total antioxidant capacity was the highest in young leaves of the dominant species *Cryptocarya chinensis* in the late-successional stage (Figure 4D). The total antioxidant capacity was significantly lower in young leaves of dominant species *Machilus chinensis* in the late-successional stage compared to *Castanopsis chinensis* in the mid-successional stage, but slightly higher compared to *Schima superba*. In general, the total antioxidant capacity was higher in young leaves of dominant species in the late-successional stage than in the mid-successional stage in both summer and winter.

### 2.5. Antioxidant Enzyme Activity

In summer and winter, young leaves of the dominant species *Schima superba* in the mid-successional stage had the highest CAT activity (Figure 5A). In summer, there was no significant difference in CAT activity between mid-successional and late-successional stages. In winter, the CAT activity was significantly higher in young leaves of the dominant species in the mid-successional stage than in the late-successional stage. Compared with summer, CAT activity in young leaves of the dominant species in both succession stages increased significantly in winter. In summer, the SOD activity was significantly higher in young leaves of dominant species in the mid-successional stage than in the late-successional stage. However, there was no significant difference in the SOD activity between the two successional stages in winter (Figure 5B).

### 2.6. Structural Equation Modeling

We analyzed the relationship between the environmental changes from summer to winter and various physiological indicators of dominant species in different successional stages. For dominant species in the mid-successional stage, environmental changes in summer and winter were directly negatively correlated with anthocyanins (−0.49) and antioxidants (−0.34), but directly positively correlated with antioxidant enzymes (0.82), and they had no effect on heat dissipation (Figure 6A). Anthocyanins had no effect on chlorophyll and Fv/Fm. Heat dissipation had no effect on chlorophyll and a direct negative effect on Fv/Fm (−0.46). Antioxidant substances (−0.87) and antioxidant enzymes (−0.49) were directly negatively correlated with Fv/Fm, and were directly negatively correlated with chlorophyll. For the dominant species in the late-successional stage, environmental changes in summer and winter were directly positively correlated with anthocyanins (0.42) and antioxidant enzymes (0.87), and directly negatively correlated with heat dissipation (−0.42), and had no effect on antioxidant substances (Figure 6B). Anthocyanins had a direct negative correlation with chlorophyll (−0.20) and no effect on Fv/Fm. Both heat dissipation and antioxidant substances were directly negatively correlated with chlorophyll and Fv/Fm. Antioxidative enzymes were directly negatively correlated with chlorophyll (−0.20) and had no effect on Fv/Fm.

## 3. Discussion

The importance of light and temperature to plant growth is obvious. However, there are differences in the adaptability of plants to different light intensities and temperatures. The differences become more obvious when they exceed the optimal growth light intensity and temperature range for plant growth. In subtropical regions, high temperature and high light are the most common environmental conditions in summer, while low temperature and high light are more common in winter. Plants grown in both environmental conditions tend to be susceptible to stress damage. The damage is mainly caused by the formation of excess light energy [34,35,36], the consequence of which is the accumulation of reactive oxygen species in the body. There are two main types of damages in plants caused by reactive oxygen species: the destruction of intracellular substances [37,38] and peroxidation of cell membranes [39,40], and the generation of photoinhibition [41,42,43,44]. In a stress environment, young leaves are more vulnerable to damage. On the one hand, the photosynthetic mechanism of young leaves is immature and the accumulated chlorophyll is less (Figure 3A), so young leaves can only use a small amount of light energy. On the other hand, young leaves generally grow in the canopy and are exposed to more light. Relative membrane leakage is often used to characterize the degree of cell membrane damage. The relative membrane leakage was higher for young leaves of the dominant species in the mid-successional stage than in the late-successional stage in both summer and winter (Figure 1). The results show more serious peroxidation of reactive oxygen species on the cell membrane of young leaves of dominant species in the mid-successional stage.

Higher membrane permeability in plants is often related to environmental stress, which can cause photoinhibition. A parameter commonly used to characterize the degree of photoinhibition is Fv/Fm [45,46]. Chlorophyll fluorescence results show that the young leaves of dominant species in the mid-successional and late-successional stages showed opposite Fv/Fm trends in different seasons, i.e., those in the mid-successional stage showed higher Fv/Fm in summer (Figure 2A). Studies have shown that the dominant species in the mid-successional stage are shade-intolerant species [47], so the higher Fv/Fm was in line with the characteristics of these species as heliophytic, while the dominant species in the late-successional stage are shade-tolerant species, and the lower Fv/Fm shows that they are not tolerant to high light. However, in a similar study, it was found that the young leaves of the dominant species in the late-successional stage had higher Fv/Fm recovery ability [48]. Compared with the mid-successional stage, young leaves of dominant species in the late-successional stage had higher Fv/Fm in winter (Figure 2A). This indicates that tolerance of the dominant species in the two stages to the two environments differed, i.e., young leaves in the mid-successional stage were more tolerant to high temperature and high light in summer, which was consistent with the conclusion of Yu et al. [48]. In winter, young leaves in the late-successional stage had stronger tolerance to low temperature and high light.

To adapt to the environmental conditions in summer and winter, plants often use a combination of defense mechanisms. Redness of young leaves is a common phenomenon in subtropical forests. However, the phenomenon of young leaves of dominant species turning red appeared in different seasons depending on the stage, i.e., summer for the mid-successional stage and winter for the late-successional stage. Quantitative determination confirmed that the appearance of red color in young leaves was due to the accumulation of anthocyanins (Figure 4A). The physiological significance of anthocyanin accumulation has been explained differently in different studies, including reactive oxygen species scavenging [49], light shielding [50], metal chelators [51], etc. In recent years, more studies have shown that anthocyanins are mainly involved in photoprotection by screening excess light energy. A previous study confirmed that anthocyanins played a light-shielding role in the mid-successional dominant species [48]. In research conducted by Zhang [52], the author speculated that the main function of anthocyanins in the dominant species in the late-successional stage is not the antioxidant effect, so we supposed that the anthocyanins in these species at this stage may also be secondary metabolites involved in light shielding. Anthocyanins in dominant species in mid- and late-successional stage functioned in summer and winter, respectively, which shows that the environmental change from summer to winter has a direct negative effect on the anthocyanins in young leaves in the mid-successional stage and a positive effect in the late-successional stage (Figure 6).

Among other secondary metabolites, the differences in flavonoid content were also extremely significant. In both summer and winter, the flavonoids content was significantly higher in young leaves of dominant species in the late-successional stage than in the mid-successional stage (Figure 4B). Flavonoids are important antioxidant substances synthesized by plants under environmental stress. In previous studies, flavonoids were shown to have stronger antioxidant properties than other antioxidants, such as ascorbic acid and tocopherol [53,54]. Therefore, flavonoids might be selected as an important part of the antioxidant system by dominant species in the late-successional stage in summer and winter, while those in the mid-successional stage may not need many flavonoids to provide photoprotection because of the light shielding of anthocyanins. In addition, compared with summer, the flavonoid content in young leaves of dominant species in the late-successional stage obviously decreased in winter. We speculated that part of the reason might be that anthocyanins synthesis consumes some of the materials for flavonoid synthesis, because the anthocyanin synthesis pathway belongs to a part of the flavonoid synthesis pathway.

Similar to flavonoids, the total phenolic content was significantly higher in young leaves of dominant species in the late-successional stage than in the mid-successional stage in both summer and winter (Figure 4C). Phenolic substances in plants are also an important part of the antioxidant system. Compared with other antioxidant substances, phenols have better antioxidant properties [21]. Interestingly, the total phenolic content in young leaves of dominant species in the late-successional stage was not significantly different in summer and winter, while the content in young leaves in the mid-successional stage was significantly lower in winter than in summer (Figure 4C). However, for individual trees, the total phenolic content of young leaves of *Machilus chinensis* was not higher than that of trees in the mid-successional stage in summer. The reason for this result is that the chlorophyll content of young leaves of *Machilus chinensis* was the highest among all trees. This same could also be observed for the total antioxidant capacity. So, we supposed that the contribution of total phenols to the antioxidant capacity of dominant species in the mid-successional stage might be greater in winter. Changes in the total antioxidant capacity could indirectly verify our guess. From summer to winter, the total antioxidant capacity of young leaves of dominant species in both mid- and late-successional stages decreased (Figure 4D). The change in total antioxidant capacity of young leaves in the mid-successional stage was consistent with the change trend of the total phenolic content.

The total antioxidant capacity of young leaves of dominant species in the late-successional stage was consistent with the change trend of the flavonoid content, which agrees with the view of Zhang et al. [55] that flavonoids play a greater role in photoprotection of dominant species in the late-successional stage. In addition, the total antioxidant capacity was significantly higher in young leaves of dominant species in the late-successional stage than in the mid-successional stage in both summer and winter (Figure 4D). Furthermore, the total phenolics content in young leaves of dominant species in the mid-successional stage and the flavonoids content and total antioxidant capacity of young leaves in both successional stages were significantly decreased in winter compared with summer (Figure 4). However, the structural equation model results show that the environmental change from summer to winter had a significant negative effect on the antioxidant substances of dominant species in the mid-successional stage, but no significant effect on those in the late-successional stage (Figure 6). We speculated that this might be affected by the higher total phenolic content of dominant species in the late-successional stage. This also indicates that the environmental changes from summer to winter have a greater impact on the antioxidant substances of the dominant species in the mid-successional stage.

An important way for plants to carry out photoprotection is to use heat dissipation to eliminate excess light energy [56,57]. In summer, NPQ was significantly higher in young leaves of dominant species in the late-successional stage than in the mid-successional stage (Figure 2B). We believe that in mid-successional stage, young leaves rely on the light-shielding of anthocyanins without accumulating excessive excitation energy, so NPQ is not very necessary for heat dissipation. In the late-successional stage, young leaves did not accumulate anthocyanins. If NPQ does not contribute to photoprotection, it means that strong total antioxidant capacity might not provide sufficient photoprotection for young leaves. Compared with summer, the NPQ of young leaves of dominant species in the late-successional stage in winter was increased, which was consistent with the results of Zhu et al. [31], indicating that in the case of insufficient photoprotection mediated by NPQ, the photoprotection ability of young leaves in the late-successional stage could be compensated by the accumulation of anthocyanins. In the mid-successional stage, the NPQ of young leaves of increased, which might serve to supplement the lack of anthocyanin and the decline in the total antioxidant capacity.

Carotenoids are important photoprotective substances in plants. Car/Chl can be used to characterize the potential to mediate the quenching of excess excitation energy. In both summer and winter, the Car/Chl was significantly higher in young leaves of the late-successional dominant species than in the mid-successional stage (Figure 3C), suggesting that NPQ mediated by the xanthophyll cycle is only a small part of the plant’s heat dissipation system, and young leaves in the late-successional stage had a stronger ability to dissipate heat than those in the mid-successional stage. On the other hand, dominant species in the mid-successional stage had only a slightly decreased Car/Chl in winter compared with summer, and structural models showed that environmental changes from summer to winter had no significant effect on heat dissipation (Figure 6A), indicating that the slight decrease in Car/Chl might offset the significant increase in NPQ. Although the environmental changes from summer to winter had a significant negative effect on the heat dissipation of dominant species in the late-successional stage (Figure 6B), the Car/Chl ratio was significantly higher compared to those in the in mid-successional stage in both summer and winter, which also shows that the dominant species in the late-successional stage have a strong heat-dissipation ability. Although there were differences in NPQ and Car/Chl between tree species at the same successional stage, there were significant differences between the two stages, even if they were not compared as a whole.

To explore the responses of the enzymatic antioxidant systems of dominant species in different successional stages to different environments in summer and winter, we measured CAT and SOD. CAT and SOD are two important antioxidant enzymes involved in reactive oxygen species scavenging in plants under stress [58,59]. The results show that there was no significant difference in the CAT of the young leaves of dominant species in the two successional stages in summer (Figure 5A), indicating that they could rely on other photoprotection mechanisms to satisfy their photoprotection needs in the high-temperature and high-light environment, so it was not necessary to increase the CAT activity to remove reactive oxygen species. In winter, the CAT activity of young leaves of dominant species in the two successional stages increased, which was consistent with the results of the structural equation model: the environmental changes from summer to winter had a significant positive effect on antioxidant enzymes (Figure 6), indicating that it is very necessary for the dominant species in both stages to resist the low temperature and high light environment in winter by increasing the CAT activity. The increased CAT in young leaves in the mid-successional stage was greater, indicating that the CAT in these leaves is more sensitive to a low-temperature and high-light environment. When there is no anthocyanin accumulation and the total antioxidant capacity is low, the young leaves of dominant species in the mid-successional stage need to improve their CAT activity. SOD activity was significantly higher in young leaves of dominant species in the mid-successional stage than that in the late-successional stage only in summer (Figure 5B). In winter, the SOD of young leaves in both successional stages increased, indicating that increasing SOD activity is also an important way for young leaves of dominant species in different successional stages to resist the winter environment. However, the activity of SOD is obviously lower than that of CAT, and we speculate that SOD might function in an auxiliary capacity.

## 4. Materials and Methods

### 4.1. Plant Materials and Growth Conditions

This study was carried out at the School of Life Sciences, South China Normal University, Guangzhou City, Guangdong Province, China (113°20′59.05″ E, 23°8′22.39″ N). Seedlings of 2 mid-successional dominant species, *Schima superba* and *Castanopsis chinensis*, and 2 late-succession dominant species, *Machilus chinensis* and *Cryptocarya chinensis*, were excavated from Dinghushan National Nature Reserve. All tree species were planted at the experimental base of South China Normal University by pot picking. The climatic conditions of the transplanted sites were consistent with those of the original planting sites, which belong to a humid monsoon climate in a low-altitude subtropical zone. The size of the planting pot was 50 cm in diameter and 35 cm in depth, and the planting soil was clay and peat (volume ratio 3:1). The watering conditions were once a day in summer and once every 3 days in winter. The plants were fertilized every 3 months. The experiment was carried out 6 years after the seedlings were planted. The summer measurements were carried out in 2019 and the winter measurements were carried out in 2021. During the experiment, the highest and lowest air temperatures were 36 °C (summer) and 10 °C (winter), and the daily average light intensity was 1000−1200 μmol m^−2^ s^−2^.

### 4.2. Relative Membrane Leakage Determination

In sunny weather, fresh young leaves and mature leaves were collected, and their surfaces were washed with ultrapure water and dried. Five 8 mm-diameter leaf discs were obtained with a hole punch. Then, the leaf discs were put into a test tube containing 10 mL of ultrapure water and left to stand for 2 h at room temperature. The conductivity of the leaves was measured using a conductivity meter. First, the conductivity of the ultrapure water after soaking the leaves for 2 h was measured and recorded as R1. Then, the test tube was put into boiling water for 30 min. After cooling to room temperature, the conductivity of the ultrapure water was measured and recorded as R2. The relative membrane leakage of the leaves was expressed in percentages as the relative conductivity (R1/R2).

### 4.3. Determination of Pigment Content

Chlorophyll content: Two 8 mm-diameter leaf discs obtained from fresh leaves were put into a 2 mL centrifuge tube and smashed into powder with a little liquid nitrogen. Then, 2 mL of 80% acetone was added to the centrifuge tube for extraction at 4 °C in the dark for 24 h. The extract was pipetted into a cuvette to measure the absorbance at wavelengths of 663 nm, 645, and 470 nm using a UV2450 spectrophotometer (Shimadzu, Tokyo, Japan). The contents of chlorophyll and carotenoid were calculated referring to the formula of Wellburn [60].

Anthocyanin content: Fresh leaves were collected and washed with ultrapure water. Avoiding the main leaf veins, five 8 mm-diameter leaf discs were obtained. The leaf discs were put into a 10 mL test tube with 4 mL of 1% HCl-methanol for extraction at 4 °C for 24 h. Anthocyanin content was determined using a UV2450 spectrophotometer (Shimadzu, Tokyo, Japan). Before the measurement, 4 mL of chloroform and 2 mL of ultrapure water were added to the extract. After fully mixing, the mixture was left to stand until the liquid layers were separated, and the upper liquid was taken for testing. The upper layer liquid was pipetted into the cuvette and the absorbance was measured at 530 nm. A standard curve was prepared using cyanidin-3-O-glucoside to calculate the anthocyanin content [18].

Flavonoid content: Fresh leaves were washed and five 8 mm-diameter leaf discs were obtained, avoiding the main veins. The leaf discs were placed in a 10 mL test tube, and 3 mL of 95% methanol was added for extraction at 4 °C for 48 h. The flavonoid determination methods referred to Heimler et al. [61], with slight modifications. Then, 0.1 mL of the extract was pipetted into a 4 mL centrifuge tube, and 2.2 mL of ultrapure water, 0.2 mL of 5% NaNO_2_, 0.3 mL of 10% AlCl_3_, and 1 mL of NaOH were added in turn. After fully mixing, the mixture was immediately poured into a cuvette, and the absorbance value was measured at 510 nm using a UV2450 spectrophotometer. A standard curve was made using catechin (25−1000 μmol L^−1^) to calculate the flavonoid content in the samples.

Total phenol content: The extraction method of total phenols was similar to that of flavonoids. The determination of total phenols referred to Ainsworth and Gillespie [62]. For this procedure, 0.5 mL of the extract, 1 mL of 10% Folin phenol, and 2 mL of 0.7 mol L^−1^ NaCO_3_ was added to the cuvette in turn, and the absorbance was measured at 765 nm. The total phenolic content in the samples was calculated using gallic acid to make standards (concentrations ranging from 50 to 250 μmol L^−1^).

### 4.4. Total Antioxidant Capacity Assessment

The extraction method of total antioxidants was similar to that of flavonoids. Referring to the method of Saha et al. [63], the total antioxidant capacity of the samples was evaluated by the scavenging ability of the extract to scavenge 1,1-diphenyl-2-picrylhydrazyl (DPPH). For this test, 20 μL of the extraction solution and 3 mL of DPPH solution were added into a 4 mL centrifuge tube, then the mixture was left to stand in the dark for 5 min to measure the absorbance at 517 nm. The DPPH scavenging ability of the samples was calculated by making a standard curve with different concentrations of DPPH solution. The total antioxidant capacity of leaves was expressed as the micromoles of DPPH scavenged per unit area.

### 4.5. Chlorophyll Fluorescence Assay

Chlorophyll fluorescence in leaves was measured using a chlorophyll fluorescence imaging system (Technologica, Colchester, UK). Fresh leaves were washed and then placed in the dark for 20 min. The leaves were first irradiated with a measurement light below 0.05 μmol m^−2^ s^−1^ to obtain the initial fluorescence (Fo), and then the maximum fluorescence (Fm) was determined with a pulsed light at 6000 μmol m^−2^ s^−1^ saturation. The maximum potential photochemical efficiency (Fv/Fm = 1 − Fo/Fm) of photosystem II (PSII) was calculated from Fo and Fm [64]. The steady-state maximum fluorescence, Fm’, was then determined under actinic light at 800 μmol m^−2^ s^−1^, and the non-photochemical quenching (NPQ = (Fm/Fm′) − 1) [65] of PSII was calculated.

### 4.6. Antioxidant Enzyme Activity Assay

After removing the main veins of fresh leaves, 0.1 g of the sample was weighed and put into a mortar. Then, 1 mL of grinding buffer (containing 50 mM PBS (pH = 7.8), 0.1 M EDTA-Na_2_, 0.1% (*v*/*v*) Triton X × 100, and 2% (*w*/*v*) polyvinypyrrolidone (PVP)) and a small amount of quartz sand were added to the mortar, and the sample was ground into a homogenate. Then homogenate was then transferred to a centrifuge tube, and 1 mL of grinding buffer was used to wash the mortar. The centrifuge tube was centrifuged at 12,000× *g* for 20 min at 4 °C.

The activity of superoxide dismutase (SOD; EC 1.15.1.1) was determined with reference to Luo et al. [66] and Fu et al. [67]. For this test, 0.1 mL of the extract was pipetted into a 4 mL centrifuge tube already containing 2.6 mL mixture (containing 0.3 mL of 130 mM Met solution, 1.7 mL of 50 mM PBS (pH = 7.8), 0.3 mL of 750 μmol L^−1^ NBT solution, and 0.3 mL of 100 μmol L^−1^ EDTA-Na_2_ solution). Finally, 0.3 mL of 20 μM riboflavin solution was added. Two negative controls and one positive control were set. In the negative control tube, the enzyme extract was replaced with 50 mm PBS (pH = 7.8) and the tube was placed in the dark. The positive control tube and the sample tube were placed under a 4000 lx fluorescent lamp to react for 15 min, and then placed in the dark to terminate the reaction. Using the negative control tube as a blank reference to zero, the absorbance values of the positive control and sample tubes were measured at 560 nm.

The peroxidase (POD; EC 1.11.1.7) activity was measured with reference to Hameed et al. [68]. For this test, 0.1 mL of enzyme extract and 2.9 mL of reaction mixture (1.875 mL of 50 mM PBS (pH = 7.0), 1 mL of 30 mM H_2_O_2_ solution, and 0.025 mL of guaiacol solution) were pipetted into a cuvette, and 50 mM PBS (pH = 7.0) was used as a blank control. Measurements were made 15 s after the reaction and then every 20 s, for a total of 9 times.

### 4.7. Data Analysis

All data are presented as mean ± standard error (SE). Data analysis and multiple comparisons were performed using SPSS 25.0 software. Statistical significance was determined using one-way analysis of variance. Duncan’s multiple comparison test was used to test the significance of differences, and the significance level was 0.05. Graphs were constructed using SigmaPlot 14.0 software. In addition, structural equation modeling (SEM) [69] was used to measure the correlations among between environmental changes from summer to winter with light shielding, heat dissipation, and antioxidant capacity. SEM analysis was performed using AMOS 22.0 software (AMOS Development Corporation, Spring House, PA, USA) software. In the graph showing individual species, *n* = 5; in the graph not showing individual species, *n* = 10.

## 5. Conclusions

Young leaves are one of the parts of a plant that are the most vulnerable to environmental influences. From summer to winter, the environment in subtropical regions changes from high temperature and high light to low temperature and high light, which means that young leaves of dominant species at different successional stages might have evolved different defense strategies in these two environments. From the results obtained, the following conclusions can be drawn:In summer, dominant species in the mid-successional stage mainly reduced the degree of photoinhibition through light shielding and enzymatic antioxidant systems to achieve photoprotection. In the late-successional stage, the dominant species improved their photoprotective ability by improving their heat dissipation ability and non-enzymatic antioxidant system, which was mainly reflected in their recovery ability after photoinhibition.In winter, the photoprotective ability of dominant species in the mid-successional stage came from improved NPQ and antioxidant enzyme activity, while those in the late-successional stage maintained low photoinhibition by a combination of light shielding, heat dissipation, and enzymatic antioxidant systems.The environmental transition from summer to winter reduced the light-shielding effect and antioxidant content of tree species in the mid-successional stage, leading to an increased degree of photoinhibition, despite a slight increase in antioxidant enzyme activity. In addition, although the environmental transition from summer to winter reduced the heat dissipation capacity and antioxidant content of tree species at the late-successional stage, their strong photoprotective ability could be maintained by increasing the light-shielding effect and antioxidant enzyme activity.

In conclusion, dominant species in the mid-successional stage have high tolerance to the high-temperature and high-light environment in summer but are not suitable for the low-temperature and high-light environment in winter. On the contrary, dominant species in the late-successional stage can adapt to both high temperature and high light in summer and low temperature and high light in winter because of their flexible light protection strategies, which also explains why species in the late-successional stage can replace species in the mid-successional stage and become the main tree species in the top communities of the subtropical forests.

## Figures and Tables

**Figure 1 ijms-23-05417-f001:**
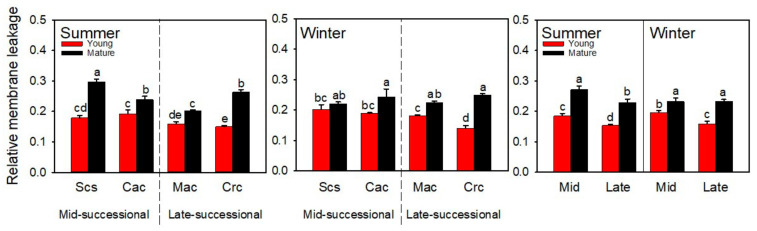
The relative membrane leakage of dominant species at different successional stages in different seasons. Scs, *Schima superba*; Cac, *Castanopsis chinensis*; Mac, *Machilus chinensis*; Crc, *Cryptocarya chinensis*; Mid, mid-successional tree group; Late, late-successional species tree group. Different letters above bars indicate statistical significance (*p* < 0.05).

**Figure 2 ijms-23-05417-f002:**
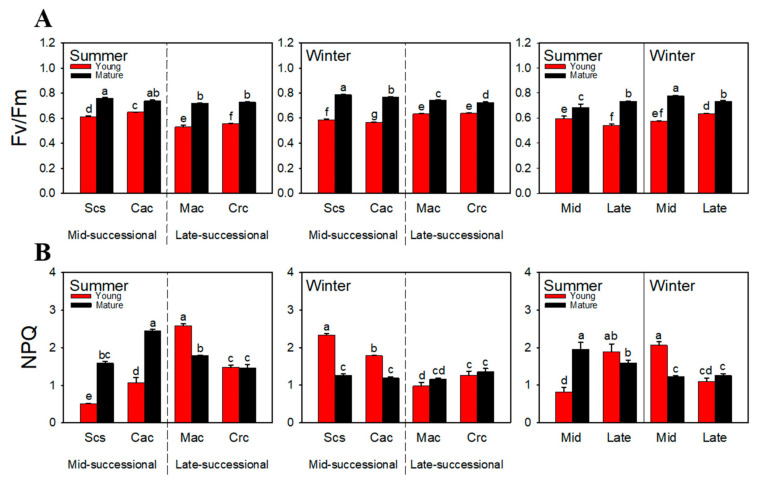
Fv/Fm (**A**) and NPQ (**B**) of dominant species at different successional stages in different seasons. Scs, *Schima superba*; Cac, *Castanopsis chinensis*; Mac, *Machilus chinensis*; Crc, *Cryptocarya chinensis*; Mid, mid-successional tree group; Late, late-successional species tree group. Different letters above bars indicate statistical significance (*p* < 0.05).

**Figure 3 ijms-23-05417-f003:**
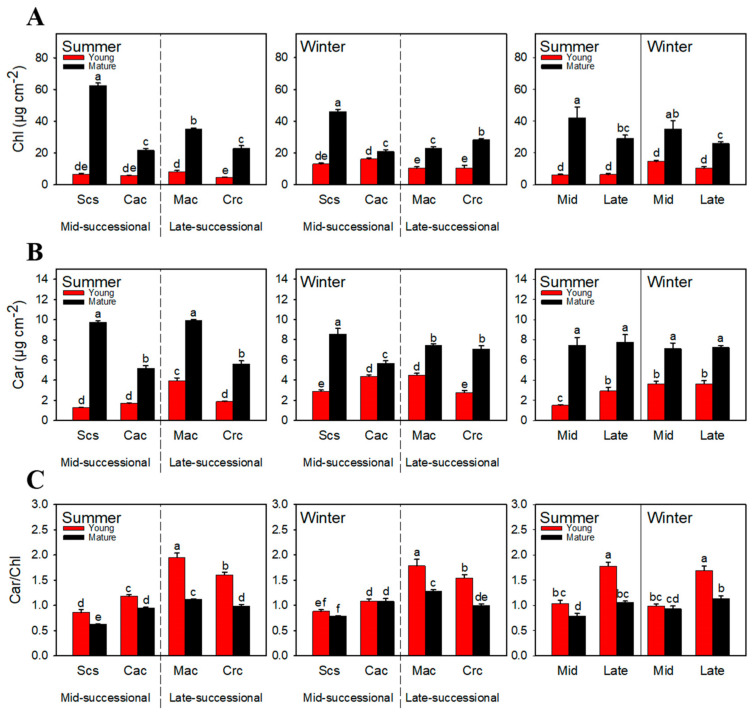
Total chlorophyll (**A**), carotenoids (**B**), and Car/Chl (**C**) of dominant species at different successional stages in different seasons. Scs, *Schima superba*; Cac, *Castanopsis chinensis*; Mac, *Machilus chinensis*; Crc, *Cryptocarya chinensis*; Mid, mid-successional tree group; Late, late-successional species tree group; Chl, total chlorophyll; Car, carotenoids. Different letters above bars indicate statistical significance (*p* < 0.05).

**Figure 4 ijms-23-05417-f004:**
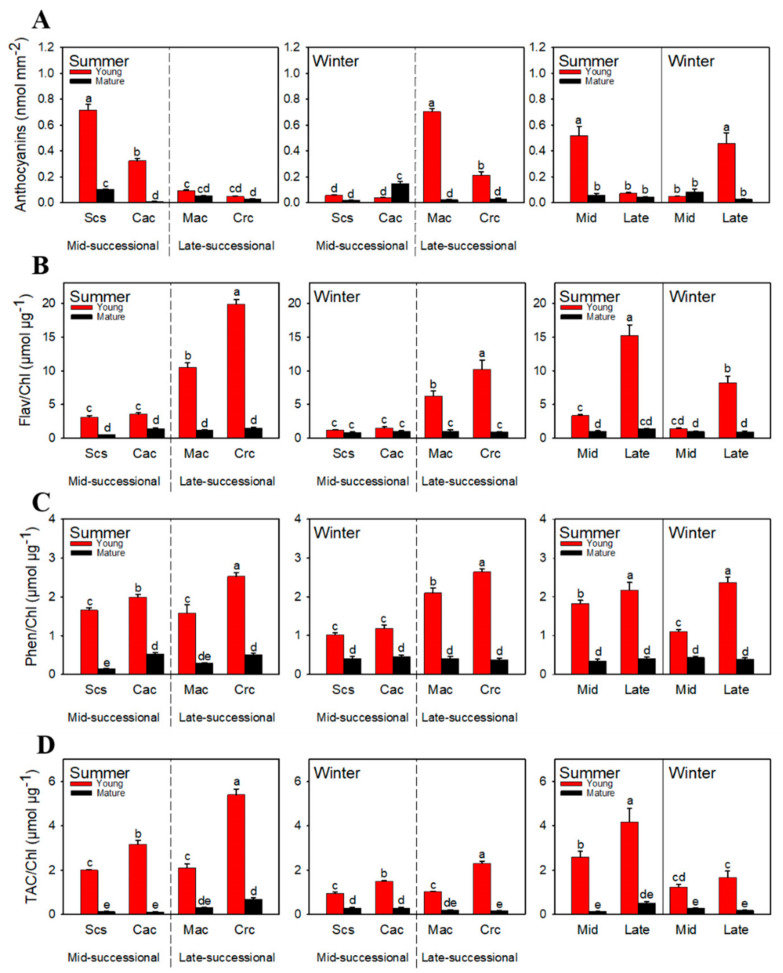
Anthocyanin (**A**), flavonoid (**B**), and total phenolic content (**C**) and total antioxidant capacity (**D**) of dominant species at different successional stages in different seasons. Scs, *Schima superba*; Cac, *Castanopsis chinensis*; Mac, *Machilus chinensis*; Crc, *Cryptocarya chinensis*; Mid, mid-successional tree group; Late, late-successional species tree group; Flav, flavonoids; Phen, phenolic; TAC, total antioxidant capacity. Different letters above bars indicate statistical significance (*p* < 0.05).

**Figure 5 ijms-23-05417-f005:**
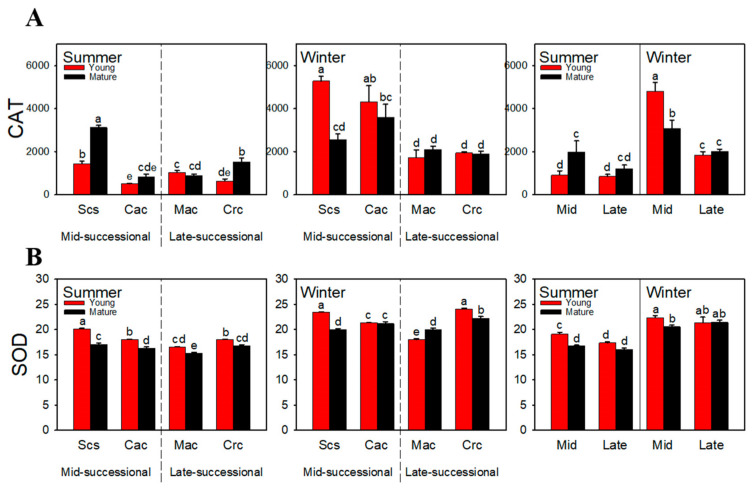
CAT (**A**) and SOD (**B**) activities of dominant species at different successional stages in different seasons. Scs, *Schima superba*; Cac, *Castanopsis chinensis*; Mac, *Machilus chinensis*; Crc, *Cryptocarya chinensis*. Mid, mid-successional tree group; Late, late-successional species tree group. Different letters above bars indicate statistical significance (*p* < 0.05).

**Figure 6 ijms-23-05417-f006:**
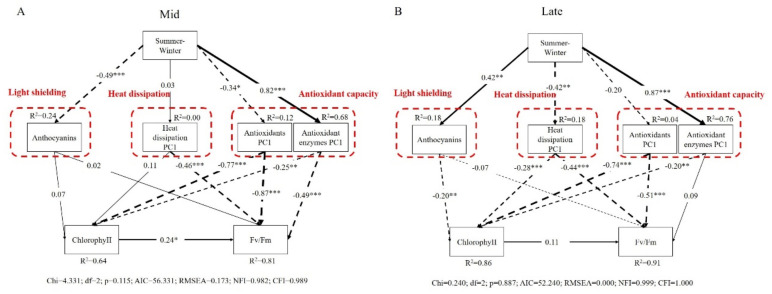
Path model of the relationships among anthocyanins, heat dissipation capacity, antioxidant substances, antioxidant enzymes, chlorophyll, and Fv/Fm of dominant species (**A**) in the mid-successional stage and late-successional stage (**B**) in summer and winter. Solid arrows represent positive effect paths and dashed arrows represent negative effect paths; thicker arrows indicate greater significance. Standardized regression coefficient is shown for each path. R2 indicates total variation in a dependent variable that is explained by the combined independent variables. Model-fit statistics are provided. Chi, Chi-square; df, degrees of freedom, p, probability level; CFI, comparative fit index; AIC, Akaike information criterion; RMSEA, root mean square error of approximation; NFI, normed fit index. * Significant difference at 0.05 level, ** 0.01 level, *** 0.001 level.

## Data Availability

Not applicable.

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
