# Peer review of "Photoprotection Differences between Dominant Tree Species at Mid- and Late-Successional Stages in Subtropical Forests in Different Seasonal Environments"

_ijms, 2022, doi:10.3390/ijms23105417_

Round 1

Reviewer 1 Report

Attached few suggestions and corrections. Authors can modify the manuscript accordingly.

The abstract and conclusion appear same.  Change the conclusion in a different pattern. Use pointwise conclusions, so that it will be essentially giving the same message in a telegraphic form.

Reviewer 2 Report

This study looked at phenolic content, oxidative potential, chlorophyll fluorescence and content, and cell membrane permeability to assess the effect of seasonal changes on young and mature leaves of dominate tree species from mid-succession and late-succession forest. The manuscript is well written and the explanation of the experimental set up and results was clear. The only major comment I have is, Do you think that the amount of light reaching young leaves is different in mid-succession and late-succession forest? Could this effect your results?

Some other minor comments.

In the graphs you state that n=5. Does n=10 in graphs that don't show individual species? i.e. all sample combined?

For some measurements, there are difference between species within a succession type. There was no discussion about these difference. Would if be better to not include these graphs?

In the first paragraph of the discussion, you state "Young leaves were more susceptible to damage be-cause they accumulated less chlorophyll (Figure 3A)." Is the cause of the susceptibility having less chlorophyll or is this a result of the stress?

Methods- Was the experiment repeated twice (once in 2019 and once in 2021) or were summer measurements done in 2019 and winter measurements done in 2021? 
